# Dietary Triple-Strain *Bacillus*-Based Probiotic Supplementation Improves Performance, Immune Function, Intestinal Morphology, and Microbial Community in Weaned Pigs

**DOI:** 10.3390/microorganisms12081536

**Published:** 2024-07-27

**Authors:** Lei Xue, Shenfei Long, Bo Cheng, Qian Song, Can Zhang, Lea Hübertz Birch Hansen, Yongshuai Sheng, Jianjun Zang, Xiangshu Piao

**Affiliations:** 1State Key Laboratory of Animal Nutrition and Feeding, College of Animal Science and Technology, China Agricultural University, Beijing 100193, China; xuelei20210228@163.com (L.X.); longshenfei@cau.edu.cn (S.L.); chengb2436426@163.com (B.C.); songqian@cau.edu.cn (Q.S.); b20233040380@cau.edu.cn (C.Z.); 2Chr. Hansen A/S, Animal and Plant Health & Nutrition, 2970 Hoersholm, Denmarkcnsesh@chr-hansen.com (Y.S.); 3Beijing Jingwa Agricultural Science and Technology Innovation Center, Beijing 101206, China

**Keywords:** *Bacillus*-based probiotic, growth performance, immunity, intestinal health, weaned pig

## Abstract

Probiotics provide health benefits and are used as feed supplements as an alternative prophylactic strategy to antibiotics. However, the effects of *Bacillus*-based probiotics containing more than two strains when supplemented to pigs are rarely elucidated. SOLVENS (SLV) is a triple-strain *Bacillus*-based probiotic. In this study, we investigate the effects of SLV on performance, immunity, intestinal morphology, and microbial community in piglets. A total of 480 weaned pigs [initial body weight (BW) of 8.13 ± 0.08 kg and 28 days of age] were assigned to three treatments in a randomized complete block design: P0: basal diet (CON); P200: CON + 200 mg SLV per kg feed (6.5 × 10^8^ CFU/kg feed); and P400: CON + 400 mg SLV per kg feed (1.3 × 10^9^ CFU/kg feed). Each treatment had 20 replicated pens with eight pigs (four male/four female) per pen. During the 31 d feeding period (Phase 1 = wean to d 14, Phase 2 = d 15 to 31 after weaning), all pigs were housed in a temperature-controlled nursery room (23 to 25 °C). Feed and water were available ad libitum. The results showed that the pigs in the P400 group increased (*p* < 0.05) average daily gain (ADG) in phase 2 and tended (*p* = 0.10) to increase ADG overall. The pigs in the P200 and P400 groups tended (*p* = 0.10) to show improved feed conversion ratios overall in comparison with control pigs. The pigs in the P200 and P400 groups increased (*p* < 0.05) serum immunoglobulin A, immunoglobulin G, and haptoglobin on d 14, and serum C-reactive protein on d 31. The pigs in the P200 group showed an increased (*p* < 0.01) villus height at the jejunum, decreased (*p* < 0.05) crypt depth at the ileum compared with other treatments, and tended (*p* = 0.09) to have an increased villus–crypt ratio at the jejunum compared with control pigs. The pigs in the P200 and P400 groups showed increased (*p* < 0.05) goblet cells in the small intestine. Moreover, the pigs in the P400 group showed down-regulated (*p* < 0.05) interleukin-4 and tumor necrosis factor-α gene expressions, whereas the pigs in the P400 group showed up-regulated occludin gene expression in the ileum. These findings suggest that SLV alleviates immunological reactions, improves intestinal microbiota balance, and reduces weaning stress in piglets. Therefore, SOLVENS has the potential to improve health and performance for piglets.

## 1. Introduction

Pigs are naturally weaned at 12–18 weeks of age, but the weaning time of pigs is often advanced to 2–4 weeks of age in the modern swine industry for economic benefits [1]. Because of an immature digestive system at weaning, separation from the sow, changes in environment and feed, as well as other factors, pigs are prone to diarrhea, depression of growth performance, and even death under weaning stress [2,3]. Diet supplementation with additives such as antibiotics has often been used to alleviate diarrhea, promote growth, and boost the immunity of nursery pigs. However, antibiotic resistance and residues have grown to be a concern for society in recent years [4]. Antibiotics have been banned for use in growth promotion in animal husbandry by policies and regulations in major animal husbandry nations and regions, including Europe, the United States, and China [5,6]. Therefore, it is of great significance to find safe, environmentally friendly, and effective alternatives to antibiotics for animal husbandry production [7]. The *Bacillus* genus possesses advantages such as acid and bile salt tolerance ability, adherence capability, anti-pathogenic activities, and is beneficial to promote growth and improve mucosal barrier function compared with other types of probiotics [8,9,10,11]. *Bacillus licheniformis* at 500 and 1000 mg/kg have been shown to improve average daily gain (ADG), antioxidant capacity, immune function, as well as regulate the intestinal microbiota and decrease the incidence of diarrhea in weaned pigs [11]. The inclusion of three different types of *Bacillus subtilis* at 1 × 10^6^ CFU/g could modulate the gut microbiota composition and metabolic activity, promote growth performance and indicators of intestinal health, as well as alleviate diarrhea [12]. Moreover, multi-strain probiotic products have been regarded as more efficacious than single-strain probiotics [13].

Little research has been carried out on the effect of *Bacillus*-based probiotics containing more than two strains when supplemented to pigs. SOLVENS (SLV) is a feed additive composed of *Bacillus subtilis* DSM 5750, *Bacillus subtilis* DSM 27273, and *Bacillus licheniformis* DSM 5749. The objectives of this study were to investigate the effects of the triple-strain *Bacillus*-based probiotic on the performance, immunity, and intestinal health of weaned pigs.

## 2. Materials and Methods

The procedures used in this study were authorized and approved by the Institutional Animal Care and Use Committee of China Agricultural University (Beijing, AW50903202-1-1). The animal experiment was conducted at the Fengning Swine Research Unit of China Agricultural University (Chengdejiuyun Agricultural and Livestock Co., Ltd., Chengde, Hebei, China).

### 2.1. Source of Probiotics

The triple-strain *Bacillus*-based probiotic (SOLVENS) used in the study was provided by Chr. Hansen A/S (Hoersholm, Denmark). The probiotic components included *Bacillus subtilis* DSM 5750, *Bacillus subtilis* DSM 27273, and *Bacillus licheniformis* DSM 5749, in the ratio of 2:1:1. The combined concentrations reached 3.25 × 10^12^ CFU/kg. The *Bacillus subtilis* strains, critical to the SLV, were isolated from soybean mash, while the *Bacillus licheniformis* strain in SLV was derived from soil.

### 2.2. Animals, Diets, and Experimental Design

A total of 480 healthy weaned pigs [initial body weight (BW) = 8.13 ± 0.08 kg, weaned at 28 days, Duroc × Landrace × Yorkshire] were randomly allocated to 3 treatments: P0: basal diet (CON); P200: CON + 200 mg SLV per kg feed (6.5 × 10^8^ CFU/kg feed); and P400: CON + 400 mg SLV per kg feed (1.3 × 10^9^ CFU/kg feed). Pigs were fed for 31 days with mash pre-starter feed from weaning to day 14 (phase 1), and mash starter feed from day 15 to 31 (phase 2) in the trial period. All experimental diets were formulated according to the National Research Council [14] to meet or exceed the nutrient requirements of pigs from 7–11 kg and 11–25 kg. The composition and nutrient levels of basal diets are shown in Table 1.

Each treatment had 20 replicated pens with 8 pigs (half males and half females) per pen. Each pen was 1.5 × 1.5 m^2^ and equipped with plastic slatted flooring, a stainless steel adjustable trough, and duckbill drinkers. The experiment was run over two rounds with 10 replicates per treatment group in each of the two rounds. The temperature and humidity in the nursery room were maintained at 23–25 °C and 50–70%, respectively. All weaned pigs had free access to feed and water throughout the trial. Daily production management and immunization, including disinfection, deworming, vaccination, etc., were performed in accordance with the management regulations of the pig farm.

### 2.3. Experimental Procedures and Sample Collection

Pigs were weighed on d 0, 14, and 31 of the experiment and the data were used to calculate ADG, average daily feed intake (ADFI), and feed conversion ratio (FCR). When the pigs dead, the weight of the dead pigs and the remaining feed were recorded repeatedly for correcting growth performance data.

The fecal score of individual pigs was recorded twice a day (10:00 h and 16:00 h). The criteria of fecal scores were based on a previous report [15]: 0 = normal, firm feces; 1 = soft feces, possible slight diarrhea; 2 = formless, semifluid feces, moderate diarrhea; 3 = very watery and frothy feces, severe diarrhea. The fecal scores ≥ 2 were defined as diarrhea. The incidence of diarrhea was calculated as follows: diarrhea incidence (%) = (total number of pigs per pen with diarrhea)/(number of pigs per pen × test period) × 100.

During the first round on the morning of d 0, 14, and 31, one pig (closest to the average BW of pigs per pen) was selected from each replicate of each treatment for collection of blood and fecal samples. Blood samples of 5 mL were drawn from the jugular vein of pigs into a vacuum tube, then centrifuged at 3000× *g* for 15 min at 4 °C. The serum was placed into 1.5 mL centrifuge tube with a pipettor and stored at −20 °C. Fresh feces were collected by massaging the rectum, placed in cryopreserved tubes, flash frozen using liquid nitrogen, and then the samples were stored at −80 °C until analysis.

On the last day of the trial in round 1, ten healthy pigs with an average BW in each treatment were sacrificed after blood collection. A 20 cm section of the ileum (10 cm close to the ileocecal junction) was isolated. Three 2 cm segments from the duodenum, the middle of the jejunum, and the isolated ileum were collected and stored in 4% paraformaldehyde at 4 °C for morphometric measurements of the small intestine. In addition, ileal mucosa scrapings were obtained by slides after rinsing the rest of the isolated ileum with sterile phosphate buffered saline and stored in liquid nitrogen for gene expression analysis.

### 2.4. Chemical Analysis of Nutrient Levels of Diets

The determination of dry matter (DM), crude protein (CP), ether extract (EE), and ash followed the methods of the Association of Official Agricultural Chemists [16]. The gross energy (GE) of diets were determined using an adiabatic oxygen bomb calorimeter (Parr 6400 Calorimeter, Parr Instruments Co., Moline, IL, USA). After treating diets with 6 mol/L HCl at 110 °C for 24 h, AAs (excluding Met, Cys, and Trp) were analyzed using an automatic amino acid analyzer (Hitachi L-8900, Hitachi Ltd., Tokyo, Japan). Met and Cys were measured as methionine sulfone and cysteic acid using an automatic amino acid analyzer (Hitachi L-8900, Hitachi Ltd., Tokyo, Japan) after oxidizing with performic acid at 0 °C for 16 h and hydrolyzing with 7.5 mol/L HCl at 110 °C for 24 h. After hydrolysis in 4 mol/L LiOH for 22 h at 110 °C, Trp was measured using high-performance liquid chromatography (Agilent 1200 Series; Aligent Technologies Inc., Santa Clara, CA, USA).

### 2.5. Blood Analysis

The concentrations of immunoglobulin A (IgA), immunoglobulin M (IgM), immunoglobulin G (IgG), C-reactive protein (CRP), and haptoglobin (Hp) were measured using enzyme-linked immunosorbent assay (ELISA) kits (Nanjing Jiancheng Bioengineering Research Institute, Nanjing, China) according to manufacturer instructions.

### 2.6. Fecal Indicators and Microbial Community Analysis

Fecal dry matter was determined according to a previously described method [16]. Fecal myeloperoxidase (MPO) and secretory immunoglobulin A (sIgA) levels were detected using the specific ELISA kits (Nanjing Jiancheng Bioengineering Research Institute, Nanjing, China).

The levels of *Escherichia coli* (*E. coli*), lactic acid bacteria (LAB), and *Clostridium perfringens* (*C. perfringens*) were determined using the dilution plate method, and microorganisms were cultured in De Man Rogosa Sharpe (MRS) medium, MacConkey Agar (MAC) medium, and Trypticase Soy Agar (TSA) medium, respectively.

### 2.7. Intestinal Morphology

The fixed intestinal tissue samples in 4% paraformaldehyde were embedded in paraffin after dehydrating, serially sliced (5 μm), and stained with hematoxylin–eosin (HE). The morphology of intestinal tissues was observed using the Nikon Eclipse E100-DS-U3 microscope imaging system (Nikon, Japan) and analyzed using Motic Med 6.0 software (Xiamen Motic Software Engineering Co., Xiamen, China). The villus height of ten intact villi and the crypt depth were measured in each slice, and five slices were selected to calculate the average value of each tissue sample. Five typical visual fields (villi complete and straight) on the slices of the duodenum, jejunum, and ileum samples were selected to measure the number of goblet cells per 100 intestinal epithelial cells.

### 2.8. Quantitative Real-Time PCR

The mRNA expressions of interleukin-1 beta (IL-1β), interleukin 4 (IL-4), interleukin 6 (IL-6), interleukin 10 (IL-10), tumor necrosis factor alpha (TNF-α), mucin 2 (MUC2), zona occludens-1 (ZO-1), claudin 1 (CLDN1), and occludin (OCDN) in the ileal mucosa were analyzed using quantitative real-time PCR (qRT-PCR). About 100 mg of each intestinal mucosal sample was added to 1 mL TRIzol reagent (Thermo Fisher Scientific, Inc., Waltham, MA, USA) and homogenized using the Precellys^®^ Evolution Homogenizer (Bead Ruptor 12, OMNI International, Inc., San Gabriel, CA, USA). The total RNA was extracted with TRIzol reagent following the manufacturer’s instructions (Invitrogen; Thermo Fisher Scientific, Inc., Waltham, USA). One μL of total RNA was used to identify the quality and purity with Nanodrop 2000 (Thermo Fisher Scientific, Inc., Waltham, USA). The extracted RNA sample was used for further analysis when the ratio of optical density 260–optical density 280 was between 1.8 and 2.0. The total RNA was reverse transcribed into cDNA through the High-Capacity cDNA Reverse Transcription Kit (Thermo Fisher Scientific, Inc., Waltham, USA) following the manufacturer’s instructions. The relative expression of RNA was analyzed using the 2^−ΔΔCT^ method and with β-actin as the reference gene. The primer sequences and product sizes for intestinal genes are listed in Table 2.

### 2.9. DNA Extraction and 16S rRNA Sequencing

Six fecal samples per treatment in each of the two rounds on d 0, d 14, and d 31 were used for 16S rRNA sequencing. Microbial genomic DNA was extracted from fecal samples using the Magnetic Soil and Stool DNA Extraction Kit (Tiangen Biochemical Technology Co., Beijing, China) following the manufacturer’s instructions. The V3-V4 region of the bacterial gene was amplified from the extracted DNA using barcode pimers 341F (5-CCTAYGGGRBGCASCAG-3) and 806R (5-GGACTACNNGGGTATCTAAT-3). PCR reactions were performed with 15 µL of Phusion^®^ High—Fidelity PCR Master Mix (New England Biolabs, Beverly, CA, USA), 2 µM of forward and reverse primers, and 10 ng template DNA. For the thermal cycling of PCR, conditions of 98 °C for 1 min for initial denaturation, 30 cycles of 98 °C for 10 s, 50 °C for 30 s, 72 °C for 30 s, and 72 °C for 5 min (final elongation) were used. The 1X loading buffer containing SYB green and PCR products were mixed and extracted from 2% agarose gel. PCR products were mixed in equal density ratios and the target bands were purified with the Universal DNA Purification Kit (Tiangen Biochemical Technology Co., Beijing, China). The NEB Next^®^ Ultra™ II FS DNA PCR-free Library Prep Kit (New England Biolabs, Beverly, USA) was used to generate sequencing libraries following the manufacturer’s instructions and the indexes were added. The check of the generated library was carried out with Qubit and real-time PCR. Quantified libraries were pooled and sequenced on the Illumina NovaSeq 6000 platforms (Illumina, San Diego, CA, USA) in accordance with the effective concentrations of the library and required amount of data.

Paired-end reads were assigned to samples according to their unique barcode and truncated by removing the barcode and primer sequence. The software FLASH (Version 1.2.11) was used to merge paired-end reads and Fastp (Version 0.23.1) was used for quality filtering of the splicing sequences. The effective tags were obtained through removing the chimera sequences by comparing the filtered splicing sequences and Silva databases (16S). For the bioinformatics analysis of the effective tags, the QIIME2 software (Version QIIME2-202006) was used.

### 2.10. Statistical Analysis

The data of growth performance, fecal microbiota, blood, and histology were verified and analyzed, and outliers were tested using the UNIVARIATE procedures of SAS (SAS Institute Inc., Cary, NC, USA). Outliers where mean values deviated from treatment means by over three times the interquartile range were identified and removed. The data of ADFI and FCR were performed using the PROC MIXED of SAS with pen as the experimental unit and other data analyses were analyzed with pig as the experimental unit. The statistical model included the main effect of diet and random effects of blocks (replicate and round). The means for each treatment were calculated using the LSMEANS statement and Tukey’s multiple range was used to separate statistical differences. *p* value ≤ 0.05 and 0.05  <  *p*  ≤ 0.10 were considered for statistical significance and tendency, respectively.

## 3. Results

### 3.1. Growth Performance and Fecal Score

The effects of SLV on the growth performance of weaned pigs are given in Table 3. There was no significant difference in BW of pigs on d 0, 14, and 31 among the three treatment groups. In comparison with P0, P400 showed higher (*p* = 0.05) ADG and lower (*p* = 0.05) ADFI in phase 2 and tended (*p* = 0.06) to have greater ADG in the overall phase (d 0 to 31). The dietary supplementation of both doses of SLV tended to decrease (*p* = 0.06) ADFI in phase 1 and improve (*p* = 0.10) the FCR in the overall period. No differences in fecal scores were observed in the first three weeks among all dietary treatments. However, pigs in P200 and P400 groups had improved (*p* = 0.03) fecal scores in week 4 compared with those in P0 group. During the trial, the diarrhea rates of the weaned pigs which were administered the three diets were not different.

### 3.2. Fecal Biomarkers and Microorganisms

According to Table 4, compared with the P0 group, the fecal dry matter (DM) content (*p* = 0.09) on d 14 in P400 and fecal sIgA levels (*p* = 0.10) on d 0 in the P200 and P400 groups tended to be higher. The fecal sIgA level on d 31 in the P200 group was lower (*p* = 0.04) than that in the P0 group. In addition, the fecal MPO level on d 14 in the P200 group was greater (*p* < 0.01) compared with the P0 and P400 groups. No significant effects on the number of *E. coli*, lactic acid bacteria, and *C. perfringens* in feces on d 0, 14, and 31 were observed among three treatments. However, only the number of lactic acid bacteria in feces at d 31 of P400 pigs was higher than that at d 0 in comparison with P0 and P400.

As presented in Table 5, the P200 and P400 groups had a lower (*p* < 0.01) relative abundance of *Streptococcus* on d 0 compared with control pigs. P200 had a significantly reduced relative abundance of *Bacteroides* on d 31 compared to P0. Additionally, P400 tended (*p* = 0.06) to increase the relative proportion of *Lactobacillus* in feces on d 14 in comparison with the P0 and P200 groups. No differences in the abundance of other bacteria genus were observed among the three treatment groups.

### 3.3. Serum Biochemical Indicators

The effects of SLV on serum biochemical indicators in weaned pigs are presented in Table 6. Compared with pigs which were fed the basal diet, the serum concentrations of IgA and IgG on d 14 were greater (*p* < 0.05) in pigs in the P200 and P400 groups (Table 6). The P200 and P400 groups tended (*p* = 0.07) to have greater IgM levels on d 14 in comparison to control pigs. In addition, the level of IgM on d 31 in the P400 group was higher (*p* = 0.04) than that in the P0 and P200 groups. The serum concentrations of CRP on d 31 was greater in pigs from the P200 and P400 groups. The P200 and P400 groups had higher (*p* < 0.01) Hp levels on d 14 in comparison to control pigs.

### 3.4. Intestinal Morphology and Immunity Parameters

As shown in Table 7, the P200 and P400 groups had a greater (*p* < 0.05) number of goblet cells in the duodenum, jejunum, and ileum compared with the P0 group. P200 pigs had a higher (*p* < 0.01) villus height in the jejunum, lower (*p* = 0.03) crypt depth in the ileum, and a tendency of increased (*p* = 0.09) villus height–crypt depth ratio (V/C) in the jejunum in comparison with control pigs. Additionally, compared to the P0 group, the mRNA expression of OCLN in the ileum of the P400 pigs was improved (*p* ≤ 0.01), whereas TNF-α and IL-4 gene expressions were reduced (*p* < 0.05). Moreover, P200 tended (*p* = 0.09) to promote IL-1β gene expression in the ileum of weaned pigs.

## 4. Discussion

Probiotics have been used widely in feed production, and several studies have highlighted the positive effects of probiotics on promoting performance and modulating intestinal microbiota after weaning [17,18]. Moreover, compared to single-strain probiotic products, mixtures of probiotic strains are suggested to be more effective [13]. In the current study, dietary supplementation with a triple-strain *Bacillus*-based probiotic (400 mg SLV per kg feed) improved ADG and ADFI from d 14 to d 31 and tended to increase ADG from d 0 to d 31. Besides, P400 tend to improve fecal DM on d 31, and P200 and P400 had a significantly decreased fecal score during week 4. Moreover, both dosages of the SLV probiotic tended to decrease FCR over the entire study. These results indicated that dietary supplementation with probiotics could promote the growth and utilization of feed and ameliorate diarrhea, which was in agreement with other publications [9,19]. Hu et al. [14] reported that pigs which were fed *Bacillus subtilis* KN-42 at 4 × 10^9^ or 20 × 10^9^ CFU/kg improved ADG and FCR, and reduced the diarrhea index of weaned pigs. The supplementation of *Bacillus subtilis* DSM 28,343 at 1 × 10^9^ CFU/kg led to higher final weight, weight gain, and significantly improved FCR in pigs [20]. Additionally, *Bacillus licheniformis* supplemented with 500 mg/kg to nursery diets significantly enhanced ADG in weaned pigs [11]. The presented results of the study also showed that the higher-dose probiotic had better effects on promoting performance than the lower-dose probiotic. As defined, probiotics conveyed a health benefit to the host when administered in adequate amounts to balance the microbial population in the gastrointestinal tract [8]. Dietary supplementation with 4 × 10^9^ or 20 × 10^9^ CFU/kg of *Bacillus subtilis* resulted in a higher ADG and G:F of pigs than 2 × 10^9^ CFU/kg *Bacillus subtilis* [14]. Moreover, the FCR of guinea pigs was improved with the increase in doses (0, 1, 2, and 3 mL) of an oral probiotic mixture [21]. On the other hand, excessive doses of probiotic supplementation may weaken the effects on promoting the growth and health of animals. The growth-promoting effects of a dose of 1 × 10^9^ CFU/d *Lactobacillus plantarum* ZJ 316 were more pronounced than those of a dose of 5 × 10^9^ CFU/d or 1 × 10^10^ CFU/d after a 60 day initial treatment on weaned pigs [22]. However, the mechanisms underlying the dose effect of probiotics on improving growth and health has not been completely elucidated and should be further investigated [9,22,23].

Secretory IgA plays a critical role in defending against enteric pathogens and is the only form of antigen-specific Ig present in the lumen of the intestine under intestinal steady-state conditions [24]. Studies have documented that sIgA contributes to shaping the gut microbiota composition and host–microbiota interactions [25,26,27]. Fecal sIgA levels can represent the concentrations of sIgA in the colon and be used as an alternative measure of immunity [28]. Dietary supplementation with *Bacillus licheniformis* at 1000 mg/kg has been shown to elevate the intestinal sIgA content of weaned pigs [10]. Moreover, late-phase laying hens fed with diets supplemented with a 500 mg/kg *Bacillus subtilis* and *Lactobacillus acidophilus* mixture powder presented higher intestinal sIgA levels compared with those fed with diets supplemented with a 250 mg/kg *Bacillus subtilis* and *Lactobacillus acidophilus* mixture powder [29]. In addition, MPO levels in feces are considered a viable biomarker for assessing inflammatory bowel disease [30,31]. In the current study, pigs fed diets supplemented with 200 mg SLV per kg feed showed lower sIgA levels on d 31 and increased MPO levels on d 14 in feces than those which were fed a basal diet, while 400 mg SLV per kg feed showed no difference compared with a basal diet. These findings indicated that the supplementation of a low dose of triple-strain *Bacillus*-based probiotics may not improve gut mucosal immunity via regulating sIgA and MPO levels, while a high dose of triple-strain *Bacillus*-based probiotics may be better for the gut mucosal immunity in this study. The mechanism of this novel finding still needs further study. Moreover, He et al. [32] demonstrated that *C. perfringens* bacteria increased sIgA levels and caused intestinal inflammation and injury in mice. Interestingly, Erdogan et al. [33] reported that MPO concentrations in mice fed with kefir (probiotics) starter culture was higher than the control group, which was related to the possible inflammation of the small intestine. In addition, no differences in MPO and sIgA levels in feces were observed in pigs which were fed diets supplemented with 400 mg SLV per kg feed and no probiotics. The histopathology of tissue samples in the small intestine and the immune system could be used to better interpret the changes [33]. However, the current research remains to be further studied.

Dietary triple-strain *Bacillus*-based probiotic supplementation at 400 mg/kg increased the number of lactic acid bacteria in feces on d 31 and tended to improve the relative abundance of *Lactobacillus* in feces on d 14. Consistently, these finding suggest that this probiotic contributes to regulating the gut microbiota of pigs. Lactic acid bacteria can benefit pigs by adjusting the gut environment, inhibiting or killing pathogens in the gastrointestinal tract, regulating the balance of intestinal flora, improving intestinal mucosal immunity, and maintaining intestinal barrier function [34]. The inclusion of *Bacillus licheniformis* at 1 × 10^9^ CFU/kg and *Clostridium butyricum* ZJU-F1 at 1 × 10^8^ CFU/kg increased the proportion of *Lactobacillus* and the levels of butyric acid, propionic acid, acetic acid, and total acid in the ceca of pigs [35]. In addition, the joint application of the *Bacillus* and *Lactobacillus* may play a cooperative role in promoting growth and improving mucosal barrier function in pigs [36,37]. Nevertheless, the detailed coordination mechanism has not been fully clarified. Mahmud et al. [38] found that *Lactobacillus* and *Bacteroides* were the dominant genera of sucking piglets and were gradually replaced by *Clostridium sensu stricto 1* with increasing age. Moreover, the relative abundance of *Lactobacillus* gradually reduced approaching the weaning period [39,40], but increased when the pigs were 8 to 9 weeks old [41]. Similarly, *Clostridium sensu stricto 1* was predominant at d 0, and the relative proportion of *Lactobacillus* in feces increased from d 0 to d 31 after weaning in this study. The possible reason for the shift in bacterial species is that fiber-rich feed after weaning promotes the increase in the abundance of *Lactobacillus*, some of which are known to degrade polysaccharides [41]. As well, the proliferation of harmful bacteria may cause an intestinal inflammatory response due to the decrease in fecal *Bacteroides* [42]. Taken together, the dietary inclusion of 400 mg SLV per kg feed could improve the microbial community of weaned pigs, and the regulatory effect is better than a supplementation of 200 mg SLV per kg feed.

In the current study, the dietary supplementation of the triple-strain *Bacillus*-based probiotic, at both tested dosages, significantly increased IgA and IgG levels in the serum. Moreover, there was a greater serum concentration of IgM at d 14 and d 31 when 400 mg SLV per kg feed was fed to nursery pigs. These results of the immunoglobulin tests indicate that the dietary inclusion of *Bacillus*-based probiotics improves the blood health indices and immune status of pigs, which was in line with previous work. Immunoglobulins are macromolecular glycoproteins mediate the adaptive immune response by identifying pathogens, preventing them from entering the host, boosting neutralization, and enabling clearance and destruction [43]. Konieczka et al. [44] added *Bacillus subtilis* and *Bacillus amyloliquefaciens* at 1.1 × 10^9^ CFU/kg to the diets of pregnant and lactating sows to explore the effects on the growth performance of sows and the intestinal development of piglets. The results showed that the dual-strain *Bacillus*-based probiotic significantly improved the serum concentrations of IgG of sows at farrowing and the serum concentration of IgM and ileal mucosa of pigs at weaning. Studies have confirmed that CRP and Hp exert immunoregulatory functions in the innate and adaptive immune system [45,46]. Both CRP and Hp are acute phase proteins and early biomarkers for inflammation stimulated by bacterial lipopolysaccharide [47]. CRP is also regarded as the non-specific marker of cytokine-induced inflammation and complement-mediated anti-inflammation [48]. During intravascular hemolysis, free hemoglobin would be released into the blood circulation, while Hp binds to the free hemoglobin (Hb) [49]. The Hp–Hb complex promotes the clearance of Hb through endocytosis by binding to CD163, the scavenger receptor on the surface of macrophages, and prevents the Hb-related toxicological effect. In the present study, the concentrations of CRP in pigs from the three treatments were consistent with reported concentrations of healthy nursery pigs and had not reached levels associated with diarrhea, catarrh, fever, or the inflammation of extremities [50]. Similarly, serum concentrations of Hp were within the range of healthy pigs for both probiotic-supplemented groups [51,52]. Moreover, no signs of inflammation or disease were evident in this study. Therefore, the inclusion of the SLV *Bacillus*-based probiotic in our study elevated serum concentrations of Hp and CRP, which was not negative to the serum immunity.

Intestinal morphology has been a common index to assess the intestinal digestive function [53]. The intestinal epithelium forms a physical barrier that defends against numerous challenges and mucus produced by goblet cells provides further protection for the intestinal tract [54]. The present results showed that pigs fed diets with the SLV probiotic at 200 mg/kg feed had a positive effect on the villus height and the V/C ratio in the jejunum, and lowered the crypt depth in the ileum. Furthermore, both dosages of SLV increased goblet cells in the duodenum and ileum, and P200 increased goblet cells in the jejunum as well. These findings indicate that the dietary supplementation of the triple-strain *Bacillus*-based probiotic can promote nutrient digestion and absorption as well as mucosal barrier and intestinal immune function. Aliakbarpour et al. [55] found that broilers fed diets with *Bacillus subtilis* at 50 and 1000 mg/kg of a mono-strain or multi-bacterial probiotic had greater growth performance after a 42 day treatment than basal diets because of the increased goblet cell number and villus length. The oral administration of a low-dose (3.9 × 10^8^ CFU/d) mixture of *Bacillus licheniformis*-*B. subtilis* improved the enteritis of weaned pigs that were supposed to be Enterotoxigenic *E. coli* F4ab/ac receptor negative [56]. The possible reasons for this were that the inclusion of *Bacillus licheniformis*-*B. subtilis* up-regulated Atoh1 expression, increased goblet cells numbers and MUC2 production, preserved the mucus barrier, and enhanced host defense capability. Additionally, in a 12-week experiment, a dose of 100 mg/kg of BW multi-strain probiotics containing a mixture of different species of lactic acid bacteria and bifidobacteria ameliorated health status, and displayed higher growth performance and meat quality of growing–finishing pigs [57]. These findings were attributed to the fact that the region-specific changes in the morphology and carbohydrate composition of mucins secreted by the intestines of pigs were induced by the multi-strain probiotics. Thus, *Bacillus*-based probiotics might help to promote nutrient digestibility, growth, and host defense against enteropathogenic bacteria by improving small intestinal morphology and increasing the number of goblet cells in the intestinal tract.

Inflammatory cytokines (IL-1β, IL-4, IL-10, TNF-α) play a role in innate and adaptive immunity and have been investigated intensively in previous studies [58,59,60]. In the current study, a dose of 400 mg SLV per kg feed down-regulated the mRNA expression of the pro-inflammatory cytokine TNF-α; the inclusion of 200 mg SLV per kg feed reduced the mRNA expression of the proinflammatory cytokine IL-1β. The findings reveal that the gene expression of proinflammatory cytokines were suppressed by the triple-strain *Bacillus*-based probiotic, which was in line with previous studies. *Bacillus amyloliquefaciens* SC06 at 2 × 10^8^ CFU/kg down-regulated the TNF-α and IL-1α transcription of piglets, but up-regulated the gene expression of IL-6 and IL-8, as well as activated the TLR signaling pathway, which had a positive effect on increasing intestinal epithelial cell barrier and immune function [61]. However, a high-dose supplementation of *Bacillus* spp.-based probiotics reduced the gene expression of anti-inflammatory cytokine IL-4 in the ileum, which may be due to the suppression of the related signaling pathway [62], yet the precise mechanism remains unclear. MUC2 mucin plays a critical role in protecting the gut barrier, regulating microbiome homeostasis, and preventing diseases [63]. However, no difference in MUC2 gene expression was observed between treatment groups in this study. Several reviews documented the vital role of tight junction proteins (ZO-1, CLDN-1, CLDN-5, OCLN and other proteins) in tight junction, barrier function, proliferation, and the apoptosis of epithelial cells [64,65]. Among these, OCLN is the only known intact membrane protein localized at the points of the membrane–membrane interaction of the tight junctions [66]. Animal experiments have shown that the overexpression of OCLD could enhance the epithelial barrier function [66,67]. Thus, the findings indicate that inclusion of 400 mg SVL per kg feed can improve intestinal barrier function via increasing the relative mRNA expression of OCLN.

## 5. Conclusions

In summary, a dietary supplementation of 400 mg/kg SOLVENS, a triple-strain *Bacillus*-based probiotic, contributed to promote performance and immune function and improve the intestinal morphology and the microbial community of weaned pigs. Moreover, the *Bacillus*-based probiotic can play a role in alleviating weaning stress and promoting the healthy status of pigs.

## Figures and Tables

**Table 1 microorganisms-12-01536-t001:** Composition and nutrient levels of basal diet.

Ingredients, %	Phase 1 (1–14 d) ^1^	Phase 2 (15–31 d)
Corn	41.66	61.73
Extruded corn	15.00	0.00
Soybean meal	14.00	19.00
Extruded full-fat soybean	12.00	7.10
Fish meal	6.00	2.00
Whey power	5.00	4.00
Soy oil	2.94	2.62
Dicalcium phosphate	0.80	0.80
Limestone	0.75	0.92
Salt	0.20	0.30
L-Lysine	0.44	0.48
Methionine	0.07	0.10
Threonine	0.15	0.16
Tryptophan	0.04	0.04
Zinc oxide	0.20	0.00
Chromic oxide (Cr_2_O_3_)	0.25	0.25
Vitamin–mineral premix ^2^	0.50	0.50
Total	100.00	100.00
Calculated composition ^3^, %		
Digestible energy, kcal/kg	3542	3490
Crude protein	19.64	18.00
Calcium	0.80	0.70
STTD Phosphorus	0.40	0.33
SID Lysine	1.35	1.23
SID Methionine	0.39	0.36
SID Threonine	0.79	0.73
SID Tryptophan	0.22	0.20
Analyzed composition %		
Gross energy, kcal/kg	4190	4018
Dry matter	90.48	89.31
Crude protein	20.28	18.08
Ether extract	6.93	5.96
Ash	5.94	5.99
Lysine	1.36	1.35
Methionine	0.37	0.37
Threonine	0.87	0.70
Tryptophan	0.26	0.25

STTD: standardized total tract digestible. SID: standardized ileal digestible. ^1^ d 1 of phase 1 was at 28 days of age. ^2^ Premix for each kg diet: vitamin A, 12,000 IU; vitamin D_3_, 2000 IU; vitamin E, 30 IU; vitamin K_3_, 30 mg; vitamin B_1_, 1.5 mg; vitamin B_6_, 3 mg; vitamin B_12_, 12 μg; riboflavin, 4 mg; pantothenic acid, 15 mg; nicotinic acid, 40 mg; choline chloride, 400 mg; folic acid, 0.7 mg; biotin, 0.1 mg; manganese,4 mg; iron, 90 mg; copper, 8.8 mg; iodine, 0.14 mg; selenium, 0.3 mg. ^3^ The values were calculated according to NRC [14].

**Table 2 microorganisms-12-01536-t002:** Primer sequences of PCR reaction.

Genes	Primer		Size (bp)	Accession Number
	D ^1^	Sequence (5′-3′)		
β-actin	F	TACGCCAACACGGTGCTGTC	207	NM_214055
	R	GTACTCTGCTTGCTGATCCACAT		
IL-1β	F	CATCAGCACCTCTCAAGCAGAACA	189	NM_214123
	R	CAGGCAGCAACCATGTACCAACT		
IL-4	F	GCCTCCTGAGCGGACTTG	122	NM_214399
	R	CTCCTTCATAATOGTCTTTAGCCT		
IL-6	F	CAGTCCAGTCGCCTTCTCCC	93	NM_214041
	R	GCATCACCTTTGGCATCTTCTT		
IL-10	F	ACCTCATTCCCCAAACACTTCA	134	NM_214022
	R	ACAAACAATGTAACATTCCCAGAG		
TNF-α	F	GCATCGCCGTCTCCTACC	201	XM_021082584
	R	GCCCAGATTCAGCAAAGTCC		
MUC2	F	CGGCTCTCCAGTCTACTCGTCTAA	204	XM_021098827
	R	TGGTTGTCGGGCAAGTTGATGA		
ZO-1	F	GAGGATGGTCACACCGTGGT	169	NM_001244539
	R	GGAGGATGCTGTTGTCTCGG		
CLDN1	F	CAAAACCTTCGCCTTCCAG	293	NM_001163647
	R	TCCCCACATTCGAGATGATTAC		
OCLN	F	ATGCTTTCTCAGCCAGCGTA	176	XM_003124280
	R	AAGGTTCCATAGCCTCGGTC		

^1^ D: direction. F: forward. R: reverse. IL: Interleukin. TNF-α, tumor necrosis factor alpha. MUC2: mucin 2. ZO-1: zona occludens-1. CLDN-1: claudin 1. OCLN: occludin.

**Table 3 microorganisms-12-01536-t003:** Effects of SLV on performance in weaned pigs.

Parameter	Treatment			SEM	*p*-Value
	P0	P200	P400		
N	160	160	160		
BW_d 0_, kg	8.13	8.13	8.13	0.08	1.00
BW_d 14_, kg	12.46	12.40	12.44	0.11	0.93
BW_d 31_, kg	19.99	20.28	20.53	0.22	0.22
Phase 1 (d 0 to 14)					
ADG, g/d	308	305	308	5.05	0.83
ADFI, g/d	474 ^x^	455 ^y^	453 ^y^	6.69	0.06
FCR, g/g	1.55	1.52	1.49	0.02	0.19
Phase 2 (d 15 to 31)					
ADG, g/d	442 ^b^	463 ^ab^	475 ^a^	9.71	0.05
ADFI, g/d	770 ^b^	774 ^b^	814 ^a^	12.79	0.04
FCR, g/g	1.76	1.69	1.73	0.03	0.24
Overall phase (d 0 to 31)					
ADG, g/d	382 ^y^	392 ^xy^	400 ^x^	5.68	0.10
ADFI, g/d	636	630	651	7.88	0.16
FCR, g/g	1.67 ^x^	1.62 ^y^	1.64 ^y^	0.02	0.10
Fecal score					
Fecal score week 1	0.25	0.31	0.30	0.05	0.67
Fecal score week 2	0.33	0.35	0.32	0.04	0.82
Fecal score week 3	0.42	0.46	0.40	0.03	0.24
Fecal score week 4	0.45 ^a^	0.38 ^b^	0.37 ^b^	0.02	0.03
Diarrhea rate, %					
Diarrhea rate week 1	1.38	1.75	1.63	0.59	0.90
Diarrhea rate week 2	1.40	0.69	1.00	0.41	0.48
Diarrhea rate week 3	1.07	0.54	0.63	0.27	0.34
Diarrhea rate week 4	0.52	0.47	0.16	0.24	0.51

All values represent the mean of 20 pens. BW: body weight. ADG: average daily gain. ADFI: average daily feed intake. FCR: feed conversion ratio. SEM: standard error of the mean. P0: basal diet with no probiotics as control group; P200: basal diet + 200 mg SLV per kg feed; P400: basal diet + 400 mg SLV per kg feed. Different superscripts in same row represent significance (a, b *p* ≤ 0.05) or tendency (x, y 0.05 < *p* ≤ 0.10).

**Table 4 microorganisms-12-01536-t004:** Effect of SLV on fecal biomarkers and microorganisms in weaned pigs.

Parameter	Treatment			SEM	*p*-Value
	P0	P200	P400		
DM of feces, %					
d 0	77.66	77.49	77.16	0.23	0.32
d 14	78.42 ^y^	78.35 ^y^	78.96 ^x^	0.20	0.09
d 31	79.44	80.02	80.04	0.22	0.12
sIgA in feces (ug/g)					
d 0	26.72 ^y^	35.32 ^x^	35.25 ^x^	3.04	0.10
d 14	46.16	37.95	55.12	7.68	0.31
d 31	64.89 ^a^	44.81 ^b^	56.98 ^ab^	5.20	0.04
MPO in feces (U/g)					
d 0	15.01	15.05	13.90	0.53	0.25
d 14	17.53 ^b^	21.14 ^a^	16.49 ^b^	0.80	<0.01
d 31	22.52	24.80	21.78	1.14	0.17
*E. coli* (10^7^ CFU/g)					
d 0	3.62	4.08	3.26	0.47	0.47
d 14	3.07	3.07	4.11	0.52	0.29
d 31	3.52	3.06	2.94	0.76	0.85
LAB (10^7^ CFU/g)					
d 0	8.24	9.81	7.74	2.15	0.78
d 14	6.84	6.70	8.25	0.98	0.48
d 31	5.68	5.02	10.67	2.59	0.27
*C. perfringens* (10^7^ CFU/g)					
d 0	2.30	2.45	2.10	0.40	0.82
d 14	2.65	2.61	3.02	0.58	0.86
d 31	1.40	3.11	1.52	0.70	0.19

All values represent the mean of 10 samples. DM: dry matter. SIgA: secretory immunoglobulin A. MPOL Myeloperoxidase. *E. coli*: Escherichia coli. LAB: *Lactic acid bacteria*. *C. Perfringens*: *Clostridium Perfringens*; SEM: standard error of the mean. P0: basal diet with no probiotics as control group; P200: basal diet + 200 mg SLV per kg feed; P400: basal diet + 400 mg SLV per kg feed. Different superscripts in same row represent significance (a, b *p* ≤ 0.05) or tendency (x, y 0.05 < *p* ≤ 0.10).

**Table 5 microorganisms-12-01536-t005:** Effects of SLV on the relative abundance of fecal microbiota of pigs at the genus levels.

Parameter	Treatment			SEM	*p*-Value
	P0	P200	P400		
d 0					
Clostridium_sensu_stricto_1	24.18	13.12	18.48	4.10	0.21
Muribaculaceae	8.72	8.79	7.13	1.79	0.76
Prevotella	5.53	5.30	7.15	1.84	0.75
Lactobacillus	2.63	3.76	0.66	1.57	0.40
Treponema	0.41	1.36	1.97	1.04	0.58
Methanobrevibacter	1.86	0.63	0.43	0.71	0.35
UCG-005	1.16	2.07	2.65	0.59	0.25
Parabacteroides	0.16	0.48	1.80	0.62	0.19
Streptococcus	4.15 ^a^	0.92 ^b^	0.91 ^b^	0.59	<0.01
Subdoligranulum	0.67	1.17	2.14	0.51	0.17
d 14					
Lactobacillus	11.95 ^x^	11.09 ^x^	22.56 ^y^	3.27	0.06
Muribaculaceae	6.14	11.45	5.37	2.10	0.13
Clostridium_sensu_stricto_1	8.52	4.54	2.10	1.95	0.11
Blautia	6.49	10.97	6.46	1.82	0.18
Prevotella	1.70	0.96	4.55	1.65	0.31
Clostridia_UCG-014	5.80	4.25	4.04	1.11	0.50
Streptococcus	1.93	2.75	3.48	1.15	0.65
Subdoligranulum	2.42	4.20	2.41	0.58	0.09
d 31					
Lactobacillus	32.99	28.17	37.01	4.00	0.33
Streptococcus	10.54	12.52	8.02	1.88	0.28
Blautia	7.78	8.40	7.62	0.89	0.81
Subdoligranulum	5.35	4.76	7.22	1.07	0.28
Faecalibacterium	4.98	3.46	5.08	0.82	0.33
Clostridium_sensu_stricto_6	0.30	2.75	0.71	0.94	0.19
Prevotellaceae_NK3B31_group	0.67	2.06	0.38	0.93	0.43
Prevotella	3.77	3.95	1.84	0.93	0.26
Muribaculaceae	3.59	3.07	2.39	1.12	0.76
Dialister	2.35	0.62	1.68	0.70	0.26
Bacteroides	1.99 ^a^	0.47 ^b^	1.29 ^ab^	0.37	0.05
Eubacterium_coprostanoligenes_group	0.67	1.11	0.74	0.27	0.47
Eubacterium_hallii_group	0.24	0.86	0.19	0.26	0.17
Clostridia_UCG-014	0.96	1.20	1.08	0.26	0.81
Escherichia-Shigella	0.11	0.27	0.10	0.10	0.44

All values represent the mean of 6 samples. SEM: standard error of the mean. P0: basal diet with no probiotics as control group; P200: basal diet + 200 mg SLV per kg feed; P400: basal diet + 400 mg SLV per kg feed. Different superscripts in same row represent significance (a, b *p* ≤ 0.05) or tendency (x, y 0.05 < *p* ≤ 0.10).

**Table 6 microorganisms-12-01536-t006:** Effects of SLV on serum biochemical indicators in weaned pigs.

Parameter	Treatment			SEM	*p*-Value
	P0	P200	P400		
IgA (ug/mL)					
d 0	802	908	852	50.53	0.35
d 14	726 ^b^	967 ^a^	926 ^a^	54.40	0.01
d 31	971	1058	1177	67.51	0.13
IgG (ug/mL)					
d 0	7947	7766	7286	229.95	0.14
d 14	7203 ^b^	8401 ^a^	8707 ^a^	335.08	0.01
d 31	8076	7736	8780	401.23	0.20
IgM (ug/mL)					
d 0	913	962	934	45.80	0.76
d 14	771 ^y^	986 ^x^	1007 ^x^	75.23	0.07
d 31	936 ^b^	980 ^b^	1194 ^a^	70.20	0.04
CRP (pg/mL)					
d 0	278	271	244	21.17	0.51
d 14	343	246	291	31.21	0.12
d 31	213 ^b^	353 ^a^	315 ^a^	34.00	0.02
Hp (ng/mL)					
d 0	482	527	509	31.22	0.61
d 14	546 ^b^	683 ^a^	654 ^a^	24.21	<0.01
d 31	695	712	707	18.47	0.80

All values represent the mean of 10 samples. IgA: immunoglobulin A. IgG: immunoglobulin G. IgM: immunoglobulin M. CRP: C-reactive protein. Hp: haptoglobin. SEM: standard error of the mean. P0: basal diet with no probiotics as control group; P200: basal diet + 200 mg SLV per kg feed; P400: basal diet + 400 mg SLV per kg feed. Different superscripts in same row represent significance (a, b *p* ≤ 0.05) or tendency (x, y 0.05 < *p* ≤ 0.10).

**Table 7 microorganisms-12-01536-t007:** Effect of SLV on intestinal morphology and immunity parameters.

Parameter	Treatment			SEM	*p*-Value
	P0	P200	P400		
Duodenum					
Villus height (μm)	510	504	495	31.48	0.94
Crypt depth (μm)	278	279	296	20.43	0.78
V/C	1.90	1.84	1.73	0.16	0.77
Goblet cells (number/μm^2^)	93.7 ^b^	136.3 ^a^	145.3 ^a^	8.41	0.04
Jejunum					
Villus height(μm)	465 ^b^	596 ^a^	464 ^b^	30.36	<0.01
Crypt depth (μm)	372	330	303	33.18	0.35
V/C	1.37 ^y^	1.97 ^x^	1.62 ^xy^	0.18	0.09
Goblet cells (number/μm^2^)	96.0 ^b^	112.3 ^b^	133.1 ^a^	6.18	<0.01
Ileum					
Villus height (μm)	474	496	490	29.70	0.86
Crypt depth (μm)	462 ^a^	317 ^b^	367 ^ab^	36.06	0.03
V/C	1.17	1.63	1.40	0.17	0.18
Goblet cells (number/μm^2^)	81.6 ^b^	128.5 ^a^	139.6 ^a^	6.95	<0.01
Gene expression in ileum					
IL-1β	0.91 ^x^	0.46 ^y^	0.65 ^xy^	0.13	0.09
IL-4	1.31 ^a^	0.88 ^ab^	0.41 ^b^	0.17	<0.01
IL-6	0.93	1.62	0.22	0.59	0.27
IL-10	0.95	1.12	0.56	0.25	0.30
TNF-α	1.08 ^a^	0.90 ^a^	0.20 ^b^	0.22	0.03
MUC2	1.18	0.97	1.05	0.19	0.75
ZO-1	0.88	0.81	0.80	0.09	0.78
CLDN-1	0.71	1.23	0.82	0.20	0.17
OCLN	0.88 ^b^	1.11 ^b^	2.17 ^a^	0.24	<0.01

All values represent the mean of 10 samples. V/C: villus height–crypt depth ratio. IL: Interleukin; TNF: tumor necrosis factor. MUC2: mucin 2. ZO-1: zona occludens-1. CLDN-1: claudin 1. OCLN: occludin. SEM: standard error of the mean. P0: basal diet with no probiotics as control group, P200: basal diet + 200 mg SLV per kg feed, P400: basal diet + 400 mg SLV per kg feed. Different superscripts in same row represent significance (a, b *p* ≤ 0.05) or tendency (x, y 0.05 < *p* ≤ 0.10).

## Data Availability

The data in the current study are available from the corresponding authors upon reasonable request.

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
