# Peer review of "Dietary Triple-Strain Bacillus-Based Probiotic Supplementation Improves Performance, Immune Function, Intestinal Morphology, and Microbial Community in Weaned Pigs"

_microorganisms, 2024, doi:10.3390/microorganisms12081536_

Round 1

Reviewer 1 Report

Comments and Suggestions for Authors

The study investigates the effects of a triple-strain Bacillus-based probiotic on the growth performance, immune function, and intestinal health of weaned pigs, finding that higher doses improved gut microbiota and immune responses and only limitedly improved growth metrics. The results suggest that such probiotics can alleviate weaning stress and enhance overall pig health. Overall, the paper is well-conceived and contributes valuable information to the field of animal nutrition and probiotic supplementation. The quality of the paper appears to be quite good, especially considering the detailed experimental design, comprehensive data analysis, and well-structured discussion of results. However, there are several punctuation and grammatical errors that affect readability that should be fixed in a minor revision.

Specific corrections include:

  1. Line 20: "elusive" should be "elucidated".

  2. Line 22: "BW" should be specified as "body weight (BW)".

  3. Line 23: "d" should be "days".

  4. Line 23: "randomly complete block design assigned" should be "randomized complete block design assigned".

  5. Line 28: Explain what is meant by "phase 2".

  6. Line 29: "tended (p = 0.10) to increase ADG in overall phase" should be "tended (p = 0.10) to increase ADG overall".

  7. Lines 29-30: "P200 and P400 tended (p = 0.10) to improve feed conversion ratio in overall phase comparison with control pigs" should be "P200 and P400 tended (p = 0.10) to improve feed conversion ratio overall compared with control pigs".

  8. Line 42: "Pigs are naturally weaned between at 12-18 weeks of age" - Remove "between".

  9. Line 50: "residue has grown to be a concern of society" should be "residues have grown to be a concern for society".

  10. Line 51: "banned on usage for growth promotion" should be "banned for use in growth promotion".

  11. Line 52: "in animal husbandry great nations and regions" should be "in major animal husbandry nations and regions".

  12. Line 55: "advantages for acid and bile salt tolerance ability" should be "advantages such as acid and bile salt tolerance".

  13. Line 58: "have shown to improve" should be "have been shown to improve".

  14. Lines 59-60: "decrease diarrhea incidence of weaned pigs" should be "decrease the incidence of diarrhea in weaned pigs".

  15. Lines 61-62: "while promote growth performance" should be "and promote growth performance".

  16. Lines 65-66: "including more than two strains" should be "containing more than two strains".

  17. Line 66: Erase comma after "SOLVENS (SLV)".

  18. Line 70: "scientific ground" should be "scientific basis".

  19. Lines 80-81: "proportioned at a ratio of 2:1:1" should be "in the ratio of 2:1:1".

  20. Lines 82-83: "the Bacillus licheniformis strain part of SLV" should be "the Bacillus licheniformis strain in SLV".

  21. Lines 85-86: "weaned at d 28" should be "weaned at 28 days".

  22. Lines 88-89: "mash pre-starter feed from wean to day 14" should be "mash pre-starter feed from weaning to day 14".

  23. Lines 128-129: "10 cm closed to the ileocecal junction" should be "10 cm close to the ileocecal junction".

  24. Line 205: "Illumina NovaSeq 6000 (San Diego, CA, USA) platforms" should be "Illumina NovaSeq 6000 platform (San Diego, CA, USA)".

  25. Line 222: "LSMENAS statement" should be "LSMEANS statement".

  26. Line 230: "tended (p = 0.06) have greater ADG" should be "tended (p = 0.06) to have greater ADG".

  27. Line 230: "in overall phase" should be "in the overall phase".

  28. Line 231: "tended to decrease (p = 0.06) ADFI in phase 1" should be "tended (p = 0.06) to decrease ADFI in phase 1".

  29. Line 232: "improve (p = 0.10) FCR" should be "improve (p = 0.10) the FCR".

  30. Line 245: "level" should be "levels".

  31. Line 245: "in P200 ..." should be "in the P200 ...".

  32. Line 246: "in P200 group" should be "in the P200 group".

  33. Line 247: "in P200 group" should be "in the P200 group".

  34. Line 248: "with P0" should be "with the P0".

  35. Lines 260-261: "P200 had a significantly reduced relative abundance of Bacteroides on d 31" should be "P200 had a significantly reduced relative abundance of Bacteroides on d 31 compared to P0".

  36. Line 262: "comparison with P0 and P200 groups" should be "in comparison with the P0 and P200 groups".

  37. Line 274: "on d 14 comparison with control pigs" should be "on d 14 compared to control pigs".

  38. Lines 277-278: "in comparison control pigs" should be "in comparison to control pigs".

  39. Table 6: Correct superscripts in the line CRP/d31.

  40. Line 286: "As illustrated in Table 7" should be "As shown in Table 7".

  41. Line 288: "in ileum" should be "in the ileum".

  42. Lines 289-290: "in jejunum comparison with ..." should be "in the jejunum compared with ...".

  43. Line 290: "in comparison with P0 group" should be "compared to the P0 group".

  44. Line 293: "in ileum of ..." should be "in the ileum of ...".

  45. Line 300: Erase dot before "Discussion".

  46. Lines 378-379: "the triple-strain Bacillus-based probiotic at both tested dosages significantly increased IgA and IgG levels in the serum" should be "the triple-strain Bacillus-based probiotic, at both tested dosages, significantly increased IgA and IgG levels in the serum".

  47. Lines 411-412: "V/C in the jejunum and lowered crypt depth" should be "V/C ratio in the jejunum and lowered the crypt depth".

  48. Lines 431-432: "increasing goblet cells numbers" should be "increasing the number of goblet cells".

Author Response

Thank you very much for your positive and constructive comments and suggestions on our manuscript.

Comment 1: Line 20:"elusive" should be "elucidated".

Response 1: Thank you for pointing this out. The effects of Bacillus-based probiotics containing more than two strains when supplemented to pigs are rarely reported. Therefore, we have revised "elusive" to "rarely elucidated" in Line 20 in the manuscript. This change makes it easier to understand the innovation of this study.

Comment 2: Line 22:"BW" should be specified as "body weight (BW)".

Response 2: Thank you for your suggestion. We agree with it and revised "BW" to "body weight (BW)" in in Line 22 in the manuscript. This makes the manuscript more specialized.

Comment 3: Line 23:"d" should be "days".

Response 3: We appreciate you for pointing this out and quite agree with your suggestion. The "d" has been modified to "days" in line 23 in our revised manuscript.

Comment 4: Line 23:"randomly complete block design assigned" should be "randomized complete block design assigned".

Response 4: Agree. We have changed "randomly complete block design assigned" to "randomized complete block design assigned" according to your suggestion in Line 23 in the manuscript for a better understanding.

Comment 5: Line 28: Explain what is meant by "phase 2".

Response 5: Thank you for pointing this out. Phase 2 was from d 15 to d 31 after weaning. It is described in Line 26.

Comment 6: Line 29:"tended (p = 0.10) to increase ADG in overall phase" should be "tended (p = 0.10) to increase ADG overall".

Response 6: Thank you for your suggestion. We agree with it and modified "tended (p = 0.10) to increase ADG in overall phase" to "tended (p = 0.10) to increase ADG overall" in Line 29 in revised manuscript. This makes it more concise and clear.

Comment 7: Lines 29-30:"P200 and P400 tended (p = 0.10) to improve feed conversion ratio in overall phase comparison with control pigs" should be "P200 and P400 tended (p = 0.10) to improve feed conversion ratio overall compared with control pigs".

Response 7: Agree. We have changed "P200 and P400 tended (p = 0.10) to improve feed conversion ratio in overall phase comparison with control pigs" to "P200 and P400 tended (p = 0.10) to improve feed conversion ratio overall compared with control pigs" in Lines 29-30 in the manuscript. This change makes for a more concise manuscript.

Comment 8: Line 42:"Pigs are naturally weaned between at 12-18 weeks of age" - Remove "between".

Response 8: Thank you for pointing this out. We have deleted "between" of the sentence in Line 42 in the manuscript.

Comment 9: Line 50:"residue has grown to be a concern of society" should be "residues have grown to be a concern for society".

Response 9: Agree. We have modified "residue has grown to be a concern of society" to "residues have grown to be a concern for society" in Line 50 in our revised manuscript.

Comment 10: Line 51:"banned on usage for growth promotion" should be "banned for use in growth promotion".

Response 10: We appreciate for your suggestion and agree with it. We have revised "banned on usage for growth promotion" to "banned for use in growth promotion" in Line 51 in the manuscript.

Comment 11: Line 52:"in animal husbandry great nations and regions" should be "in major animal husbandry nations and regions".

Response 11: Thank you for pointing this out. We quite agree with the opinion and changed "in animal husbandry great nations and regions" to "in major animal husbandry nations and regions" in Line 52 in the revised manuscript.

Comment 12: Line 55:"advantages for acid and bile salt tolerance ability" should be "advantages such as acid and bile salt tolerance".

Response 12: Agree. We have modified "advantages for acid and bile salt tolerance ability" to "advantages such as acid and bile salt tolerance" in Line 55 in revised manuscript.

Comment 13: Line 58:"have shown to improve" should be "have been shown to improve".

Response 13: We appreciate for your suggestion and agree with it. We have revised "have shown to improve" to "have been shown to improve" in Line 58 in the manuscript.

Comment 14: Lines 59-60:"decrease diarrhea incidence of weaned pigs" should be "decrease the incidence of diarrhea in weaned pigs".

Response 14: Thank you for pointing this out. We quite agree with the opinion and changed "decrease diarrhea incidence of weaned pigs" to "decrease the incidence of diarrhea in weaned pigs" in Lines 59-60 in the revised manuscript.

Comment 15: Lines 61-62:"while promote growth performance" should be "and promote growth performance".

Response 15: Agree. We have modified "while promote growth performance" to "and promote growth performance" in Lines 61-62 in the manuscript.

Comment 16: Lines 65-66:"including more than two strains" should be "containing more than two strains".

Response 16: We appreciate for your suggestion and agree with it. We have revised "including more than two strains" should be "containing more than two strains" in Lines 65-66 in the revised manuscript and the rest of the manuscript.

Comment 17: Line 66: Erase comma after "SOLVENS (SLV)".

Response 17: Thank you for pointing this out. We must apologize for our negligence and have erased comma after "SOLVENS (SLV)" in Line 66 in the manuscript.

Comment 18: Line 70:"scientific ground" should be "scientific basis".

Response 18: Agree. We have modified "scientific ground" to "scientific basis" in Line 70 in the revised manuscript.

Comment 19: Lines 80-81:"proportioned at a ratio of 2:1:1" should be "in the ratio of 2:1:1".

Response 19: We appreciate for your suggestion and agree with it. We have revised "proportioned at a ratio of 2:1:1" to "in the ratio of 2:1:1" in Lines 80-81 in the manuscript.

Comment 20: Lines 82-83:"the Bacillus licheniformis strain part of SLV" should be "the Bacillus licheniformis strain in SLV".

Response 20: Thank you for pointing this out. We quite agree with the opinion and changed "the Bacillus licheniformis strain part of SLV" to "the Bacillus licheniformis strain in SLV" in Lines 82-83 in the revised manuscript.

Comment 21: Lines 85-86:"weaned at d 28" should be "weaned at 28 days".

Response 21: Agree. We have modified "weaned at d 28" to "weaned at 28 days" in Lines 85-86 in the manuscript.

Comment 22: Lines 88-89:"mash pre-starter feed from wean to day 14" should be "mash pre-starter feed from weaning to day 14".

Response 22: We appreciate for your suggestion and agree with it. We have revised "mash pre-starter feed from wean to day 14" to "mash pre-starter feed from weaning to day 14" in Lines 88-89 in the revised manuscript.

Comment 23: Lines 128-129:"10 cm closed to the ileocecal junction" should be "10 cm close to the ileocecal junction".

Response 23: Thank you for pointing this out. We quite agree with the opinion and changed "10 cm closed to the ileocecal junction" to "10 cm close to the ileocecal junction" in Lines 128-129 in the manuscript.

Comment 24: Line 205:"Illumina NovaSeq 6000 (San Diego, CA, USA) platforms" should be "Illumina NovaSeq 6000 platform (San Diego, CA, USA)".

Response 24: Agree. We have modified "Illumina NovaSeq 6000 (San Diego, CA, USA) platforms" to "Illumina NovaSeq 6000 platform (San Diego, CA, USA)" in Line 205 in the revised manuscript.

Comment 25: Line 222:"LSMENAS statement" should be "LSMEANS statement".

Response 25: Thank you for pointing this out. We must apologize for our negligence and have revised "LSMENAS statement" to "LSMEANS statement" in Line 222 in the manuscript.

Comment 26: Line 230:"tended (p = 0.06) have greater ADG" should be "tended (p = 0.06) to have greater ADG".

Response 26: Thank you for your suggestion. We quite agree with it and changed "tended (p = 0.06) have greater ADG" to "tended (p = 0.06) to have greater ADG" in Line 230 in the revised manuscript.

Comment 27: Line 230:"in overall phase" should be "in the overall phase".

Response 27: Agree. We have modified "in overall phase" to "in the overall phase" in Line 230 in the manuscript.

Comment 28: Line 231:"tended to decrease (p = 0.06) ADFI in phase 1" should be "tended (p = 0.06) to decrease ADFI in phase 1".

Response 28: Thank you for pointing this out. We have revised "tended to decrease (p = 0.06) ADFI in phase 1" to "tended (p = 0.06) to decrease ADFI in phase 1" in Line 231 in the revised manuscript.

Comment 29: Line 232:"improve (p = 0.10) FCR" should be "improve (p = 0.10) the FCR".

Response 29: Thank you for your suggestion. We quite agree with it and changed "improve (p = 0.10) FCR" to "improve (p = 0.10) the FCR" in Line 232 in the manuscript.

Comment 30: Line 245:"level" should be "levels".

Response 30: Agree. We have modified "level" to "levels" in Line 245 in the revised manuscript.

Comment 31: Line 245:"in P200 ..." should be "in the P200 ...".

Response 31: Thank you for pointing this out. We agree with the opinion and have revised "in P200 ..." to "in the P200 ..." in Line 245 in the manuscript.

Comment 32: Line 246:"in P200 group" should be "in the P200 group".

Response 32: Thank you for your suggestion. We quite agree with it and changed "in P200 group" to "in the P200 group" in Line 246 in the revised manuscript.

Comment 33: Line 247:"in P200 group" should be "in the P200 group".

Response 33: Agree. We have modified "in P200 group" to "in the P200 group" in Line 247 in the manuscript.

Comment 34: Line 248:"with P0" should be "with the P0".

Response 34: Thank you for pointing this out. We agree with the opinion and have revised "with P0" to "with the P0" in Line 248 in the manuscript.

Comment 35: Lines 260-261:"P200 had a significantly reduced relative abundance of Bacteroides on d 31" should be "P200 had a significantly reduced relative abundance of Bacteroides on d 31 compared to P0".

Response 35: Thank you for your suggestion. We quite agree with it and changed "P200 had a significantly reduced relative abundance of Bacteroides on d 31" to "P200 had a significantly reduced relative abundance of Bacteroides on d 31 compared to P0" in Lines 260-261 in the revised manuscript.

Comment 36: Line 262:"comparison with P0 and P200 groups" should be "in comparison with the P0 and P200 groups".

Response 36: Agree. We have modified "comparison with P0 and P200 groups" to "in comparison with the P0 and P200 groups" in Line 262 in the manuscript.

Comment 37: Line 274:"on d 14 comparison with control pigs" should be "on d 14 compared to control pigs".

Response 37: Thank you for pointing this out. We agree with the opinion and have revised "on d 14 comparison with control pigs" to "on d 14 compared to control pigs" in Line 274 in the revised manuscript.

Comment 38: Lines 277-278:"in comparison control pigs" should be "in comparison to control pigs".

Response 38: Thank you for your suggestion. We quite agree with it and changed "in comparison control pigs" to "in comparison to control pigs" in Lines 277-278 in the revised manuscript.

Comment 39: Table 6:Correct superscripts in the line CRP/d31.

Response 39: Thank you for pointing this out. We must apologize for our negligence and have corrected superscripts in the line CRP/d31 in Table 6 in the manuscript.

Comment 40: Line 286:"As illustrated in Table 7" should be "As shown in Table 7".

Response 40: Agree. We have modified "As illustrated in Table 7" to "As shown in Table 7" in Line 286 in the manuscript.

Comment 41: Line 288:"in ileum" should be "in the ileum".

Response 41: Thank you for your suggestion. We quite agree with it and changed "in ileum" to "in the ileum in Line 288 in the revised manuscript.

Comment 42: Lines 289-290:"in jejunum comparison with ..." should be "in the jejunum compared with ...".

Response 42: Thank you for pointing this out. We agree with the opinion and have revised "in jejunum comparison with ..." to "in the jejunum compared with ..." in Lines 289-290 in the manuscript.

Comment 43: Line 290:"in comparison with P0 group" should be "compared to the P0 group".

Response 43: Agree. We have modified "in comparison with P0 group" to "compared to the P0 group" in Line 290 in the manuscript.

Comment 44: Line 293:"in ileum of ..." should be "in the ileum of ...".

Response 44: Thank you for your suggestion. We quite agree it and changed "in ileum of ..." to "in the ileum of ..." Line 293 in the revised manuscript.

Comment 45: Line 300:Erase dot before "Discussion".

Response 45: Thank you for pointing this out. We are sorry for our carelessness and have erases dot before "Discussion" in Line 300 in the manuscript.

Comment 46: Lines 378-379:"the triple-strain Bacillus-based probiotic at both tested dosages significantly increased IgA and IgG levels in the serum" should be "the triple-strain Bacillus-based probiotic, at both tested dosages, significantly increased IgA and IgG levels in the serum".

Response 46: Agree. We have modified "the triple-strain Bacillus-based probiotic at both tested dosages significantly increased IgA and IgG levels in the serum" to "the triple-strain Bacillus-based probiotic, at both tested dosages, significantly increased IgA and IgG levels in the serum" in Lines 378-379 in the manuscript.

Comment 47: Lines 411-412:"V/C in the jejunum and lowered crypt depth" should be "V/C ratio in the jejunum and lowered the crypt depth".

Response 47: Thank you for your suggestion. We quite agree with it and changed "V/C in the jejunum and lowered crypt depth" to "V/C ratio in the jejunum and lowered the crypt depth" Lines 411-412 in the revised manuscript.

Comment 48: Lines 431-432:"increasing goblet cells numbers" should be "increasing the number of goblet cells".

Response 48: Thank you for pointing this out. We have revised "increasing goblet cells numbers" to "increasing the number of goblet cells" in Lines 431-432 in the manuscript.

Reviewer 2 Report

Comments and Suggestions for Authors

The article effectively addresses multiple aspects of the study, including growth performance, immune function, intestinal morphology, and microbial community, offering a comprehensive view of the effects of Bacillus-based probiotics on weaned pigs. By exploring alternatives to antibiotics, it addresses the critical issue of antibiotic resistance. The investigation of a triple-strain Bacillus-based probiotic, a relatively novel approach compared to single-strain studies, adds significant value to the research. The authors present the data coherently, explaining the effects of different probiotic doses on growth, intestinal health, and immune responses in pigs. However, the findings regarding the impact of probiotics on sIgA and MPO levels are inconsistent, particularly with the 200 mg dose showing effects not observed at 400 mg. This inconsistency needs further clarification or exploration to understand the underlying reasons. It would be beneficial to investigate whether other studies have reported similar results and to explore potential mechanisms behind these discrepancies .

Author Response

Thank you for your insightful comments and suggestions.

Comments: The article effectively addresses multiple aspects of the study, including growth performance, immune function, intestinal morphology, and microbial community, offering a comprehensive view of the effects of Bacillus-based probiotics on weaned pigs. By exploring alternatives to antibiotics, it addresses the critical issue of antibiotic resistance. The investigation of a triple-strain Bacillus-based probiotic, a relatively novel approach compared to single-strain studies, adds significant value to the research. The authors present the data coherently, explaining the effects of different probiotic doses on growth, intestinal health, and immune responses in pigs. However, the findings regarding the impact of probiotics on sIgA and MPO levels are inconsistent, particularly with the 200 mg dose showing effects not observed at 400 mg. This inconsistency needs further clarification or exploration to understand the underlying reasons. It would be beneficial to investigate whether other studies have reported similar results and to explore potential mechanisms behind these discrepancies.

Response: Thank you very much for your comment and pointing this out. We quite agree with this comment. Therefore, we investigated other studies and found similar results. Then, we supplemented the relevant discussion content in Line 346-350 in the revised manuscript: Late-phase laying hens fed with diets supplemented with 500 mg/kg Bacillus subtilis and Lactobacillus acidophilus mixture powder presented higher intestinal sIgA levels compared with those fed with diets supplemented with 250 mg/kg Bacillus subtilis and Lactobacillus acidophilus mixture powder. However, the mechanism of inconsistent effects of different doses has not been reported and remains to be further explored.

Reviewer 3 Report

Comments and Suggestions for Authors

The study reported in this submission is (very) well conducted and involves lot of useful measurements. The paper is written well and easy to understand.

But there are some issues

 First, this is primarily a commercial trial testing and promoting a commercial probiotic product (SOLVENS [SLV]). As such, there is not much of novelty. Authors ‘claim’ that ‘Little research has been carried out on the effect of Bacillus-based probiotics including more than two strains when supplemented to pigs’ – see L65. This is not correct, because most commercial probiotics are of multi-strain. In L63-4, it is acknowledged that ‘multi-strain probiotic products are more efficacious than single-strains’. Perhaps some novelty comes from the use of defined bacillus products. Nevertheless, the data generated are useful additions to the existing literature. A specific comment is needed on the novelty claimed.

 Table 1: clearly indicate, as a footnote, that d1 in Phase 1 refers to d28. Also clarify elsewhere in the text

 No antibiotic control used in the basal diet. Please comment. Comparison with antibiotics would have been instructive.

 L215: Statistics: ‘orthogonal polynomials’ may be the appropriate model for this dataset evaluating gradual inclusions of a product. Comment.

 Although the probiotic influenced many physiological and blood parameters, it had no effect on growth or fecal scores (during the vulnerable period). Comment on the economics of additional cost.

 L457: what dose?

L70 & L461-2: redundant. Delete.

Author Response

Thank you for your valuable and professional comments and suggestions.

Comment 1: First, this is primarily a commercial trial testing and promoting a commercial probiotic product (SOLVENS [SLV]). As such, there is not much of novelty. Authors ‘claim’ that ‘Little research has been carried out on the effect of Bacillus-based probiotics including more than two strains when supplemented to pigs’ – see L65. This is not correct, because most commercial probiotics are of multi-strain. In L63-4, it is acknowledged that ‘multi-strain probiotic products are more efficacious than single-strains’. Perhaps some novelty comes from the use of defined bacillus products. Nevertheless, the data generated are useful additions to the existing literature. A specific comment is needed on the novelty claimed.

Response 1: Thank you for pointing this out. Indeed, some studies reported the application effects of compound probiotics in weaned piglets. However, the majority of research of compound parobiotics contained not only Bacillus, but also other bacteria, such as Lactobacillus, Clostridium butyricum, Bifidobacterium. The effects of probiotics based only more than two Bacillus strains were rarely reported. The Bacillus genus possesses advantages for acid and bile salt tolerance ability, adherence capability and anti-pathogenic activities, and is beneficial to promote growth and improve mucosal barrier function compared with other types of probiotics [1-3]. Therefore, we believe that the innovation lies in investigating the effects of more than two Bacillus strains in weaned pigs, but not the effects of mixture of Bacillus and other bacteria in weaned pigs.

Reference

  1. Hu, Y.; Du, Y.; Li, S.; Zhao, S.; Peng, N.; Liang Y. Effects of Bacillus subtilisKN-42 on growth performance, diarrhea and faecal bacterial flora of weaned piglets. Asian-Australas. J. Anim. Sci. 2014, 27, 1131-1140. https://doi.org/10.5713/ajas.2013.13737
  2. Sun,W.; Chen, W.; Meng, K.; Cai, L.; Li, G.; Li, X.; Jang,  Dietary supplementation with probiotic Bacillus licheniformis S6 improves intestinal integrity via modulating intestinal barrier function and microbial diversity in weaned piglets. Biology. 2023, 12, 238. https://doi.org/10.3390/biology12020238
  3. Yu, X.; Cui, Z.; Qin, S.; Zhang, R.; Wu, Y.; Liu, J.; Yang, C. Effects of Bacillus licheniformison growth performance, diarrhea incidence, antioxidant capacity, immune function, and fecal microflora in weaned piglets. Animals. 2022, 12, 1069. https://doi.org/10.3390/ani12131609

Comment 2: Table 1: clearly indicate, as a footnote, that d 1 in Phase 1 refers to d 28. Also clarify elsewhere in the text.

Response 2: Agree. We have supplemented the footnote and clarified the point according to your comment in line 98-99 in revised manuscript. This supplement is conducive to a better understanding of the age of piglets used.

Comment 3: No antibiotic control used in the basal diet. Please comment. Comparison with antibiotics would have been instructive.

Response 3: We appreciate your insightful suggestion and thank you for pointing this out. We focused on the effects of different doses compared with no probiotic in this study. However, it would be more instructive and useful if we add another treatment with antibiotics. Therefore, we would increase this treatment group in subsequent trials.

Comment 4: L215: Statistics: ‘orthogonal polynomials’ may be the appropriate model for this dataset evaluating gradual inclusions of a product. Comment.

Response 4. Thank you for your suggestion and it is pretty well. In our opinion, there were not many interaction effects in this study. However, we will use ‘orthogonal polynomials’ in the statistics if there are more interaction effects in future studies.

Comment 5: Although the probiotic influenced many physiological and blood parameters, it had no effect on growth or fecal scores (during the vulnerable period). Comment on the economics of additional cost.

Response 5: We thank your insightful comment. First, the Bacillus subtilis strains, critical to the SLV, were isolated from soybean mash, while the Bacillus licheniformis strain in SLV was derived from soil. Therefore, the probiotic is easier to prepare than other multi-strain probiotics. Although it has not been used in feed market, the price of the probiotic provided by Chr. Hansen A/S is about 9.3 CHF/kg and is relative low compared with other probiotics. The probiotic tended to improve the ADFI of weaned pigs during the vulnerable period. Next, the probiotic significantly increased ADG and improved fecal score in phase 2. We assume that the effects are also due to the supplementation of probiotics in phase 1. It is also possible that this improvement is first reflected in physiological and blood parameters and then in growth and fecal scores. Regardless, the economics are very valuable for the application of the probiotic in production and would be taken into account in our next research.

Comment 6: L457: what dose?

Response 6: Thank you for pointing this out. The dose was 400 mg / kg and we supplemented the dose in Line 457 in revised manuscript. The change helps to understand the effects of different doses of probiotics.

Comment 7:L70 & L461-2: redundant. Delete.

Response 7: Thank you for your suggestion. We have deleted the Line70 & Line 461-2 in revised manuscript. This contributes to the manuscript more academic.